# New Occurrences of the Tiger Shark (*Galeocerdo cuvier)* (Carcharhinidae) off the Coast of Rio de Janeiro, Southeastern Brazil: Seasonality Indications

**DOI:** 10.3390/ani12202774

**Published:** 2022-10-14

**Authors:** Izar Aximoff, Rodrigo Cumplido, Marcelo Tardelli Rodrigues, Ubirajara Gonçalves de Melo, Eduardo Barros Fagundes Netto, Sérgio Ricardo Santos, Rachel Ann Hauser-Davis

**Affiliations:** 1Laboratório de Radioecologia e Mudanças Globais, Núcleo de Fotografia Cientifica Ambiental, Instituto de Biologia Roberto Alcântara Gomes, Universidade do Estado do Rio de Janeiro (UERJ), Rua São Francisco Xavier, nº 524, PHLC Subsolo, Maracanã, Rio de Janeiro 20550-900, RJ, Brazil; 2Programa de Pós-Graduação em Oceanografia (PPG-OCN), Centro de Tecnologia e Ciências, Faculdade de Oceanografia, Universidade do Estado do Rio de Janeiro (UERJ), Rua São Francisco Xavier, nº 524, Maracanã, Rio de Janeiro 20550-000, RJ, Brazil; 3Laboratório de Ecotoxicologia e Microbiologia Ambiental (LEMAM), Instituto Federal de Educação, Ciência e Tecnologia Fluminense (IFF), Campus Cabo Frio, Estrada Cabo Frio-Búzios, s/nº, Baía Formosa, Cabo Frio 28909-971, RJ, Brazil; 4Programa Associado de Pós-Graduação em Biotecnologia Marinha (PPGBM), Instituto de Estudos do Mar Almirante Paulo Moreira (IEAPM), Universidade Federal Fluminense (UFF), Rua Kioto, nº 253, Praia dos Anjos, Arraial do Cabo 28930-000, RJ, Brazil; 5Divisão de Oceanografia Biológica, Departamento de Oceanografia, Instituto de Estudos do Mar Almirante Paulo Moreira (IEAPM), Rua Kioto, nº 253, Praia dos Anjos, Arraial do Cabo 28930-000, RJ, Brazil; 6Laboratório de Biologia e Tecnologia Pesqueira (BioTecPesca), Universidade Federal do Rio de Janeiro (UFRJ), Rio de Janeiro 21941-590, RJ, Brazil; 7Instituto Museu Aquário Marinho do Rio de Janeiro (IMAM-AquaRio), Rio de Janeiro 20220-360, RJ, Brazil; 8Laboratório de Avaliação e Promoção da Saúde Ambiental, Instituto Oswaldo Cruz, Fiocruz, Avenida Brasil, nº 4.365, Manguinhos, Rio de Janeiro 21040-360, RJ, Brazil

**Keywords:** elasmobranchs, geographic distribution, citizen science, southeastern Brazil

## Abstract

**Simple Summary:**

There is a lack of detailed information on the capture pressures and records regarding the tiger shark *Galeocerdo cuvier* (Péron & Lesueur, 1822) (Carcharhinidae) for the state of Rio de Janeiro, Brazil. This study aimed to expand the tiger shark record database and to improve upon future conservation and management strategies in this area. A total of 23 new records were obtained, increasing the number of tiger shark records off the coast of the state of Rio de Janeiro approximately 5-fold. Possible seasonality patterns concerning tiger shark sizes were noted, indicating the need to consider the coast of Rio de Janeiro as an especially relevant area for at least part of the life history of tiger sharks.

**Abstract:**

The tiger shark *Galeocerdo cuvier* (Péron & Lesueur, 1822) (Carcharhinidae) is classified as near-threatened along the Brazilian coast, in line with its global categorization. Although Rio de Janeiro, located in southeastern Brazil, is internationally identified as a priority shark conservation area, many shark species, including tiger sharks, are landed by both industrial and artisanal fisheries in this state. However, there is a lack of detailed information on the species capture pressures and records for the state of Rio de Janeiro. Therefore, the aims of this study were to expand the tiger shark record database and to improve upon future conservation and management strategies. Tiger shark records from four coastal Rio de Janeiro regions were obtained by direct observation. The information obtained from fishery colonies/associations, environmental guards, researchers, and scientific articles, totaling 23 records, resulted in an approximately 5-fold increase in the number of tiger shark records off the coast of the state of Rio de Janeiro. A possible seasonality pattern concerning the size of the captured/observed animals was noted, emphasizing the need to consider the coast of Rio de Janeiro as an especially relevant area for at least part of the life history of tiger sharks.

## 1. Introduction

Elasmobranchs generally play important roles in shallow-water ecosystems in reefs, bays, and estuaries in tropical and temperate oceans [1]. However, despite their environmental importance, the global abundance of oceanic sharks and rays has declined by 71% since 1970 due to an 18-fold increase in relative fishing pressure [2]. In addition, other pressures associated with climate change and chemical contamination of the oceans are also significant negative stressors to this taxonomic group [3,4]. In the long term, these pressures jeopardize populations, a problem compounded by the fact that sharks and rays are a long-lived and a late-maturing species, thus increasing the risk of global extinction to the point that three-fourths of all elasmobranch species are threatened with extinction [2].

One of the priority areas concerning global elasmobranch conservation is the Atlantic coast of South America, which presents the highest concentration of threatened species [5] and where scientific data and resources are limited, with the complexity of local fisheries often being overlooked [6]. In Brazil, the monitoring of fisheries ceased from 2011, which had landed more than 20 tons of cartilaginous fish in the previous decade [7]. It is estimated that Brazil currently ranks among the countries with the highest elasmobranch capture rates worldwide [8]. While battling finning has exposed the global fin market to public scrutiny and control agencies [9,10], the emphasis given to Brazil is validated by the increased elasmobranch meat consumption and non-fin products noted in recent decades [11]. Despite the commercial importance of elasmobranchs, the reconstruction of fishery production data indicates a considerable drop in shark landings, from a peak in the early 1980s declining in the following years to the current estimates of around 9.000 tons/year [12].

The pressure and lack of information associated with the overfishing of species traded without consumers knowing they are buying and consuming shark meat have put these animals at significant risk in Brazil [13]. Currently, one-third of Brazilian shark species are threatened [14], although the true status of many species is unknown due to lack of information [13]. This is true for the tiger shark *Galeocerdo cuvier* (Péron & Lesueur, 1822) (Carcharhinidae), which, although one of the least representative species in northeast [15], southeast [16], and south [17] Brazil, has been classified as near threatened along the Brazilian coast, in line with its global categorization [14,18]. Australia has, in fact, recently identified declines in this species abundance [19]. In southeastern Brazil, although the state of Rio de Janeiro is internationally identified as a priority shark conservation area [20], it is responsible for 2.5% of the total number of sharks caught per year in Brazil [21].

Among the several shark species caught by artisanal fisher colonies in Rio de Janeiro, the tiger shark has always been one of the least recorded [16,22,23]. The tiger shark is a large iconic predator that can reach 500 cm in length and weigh around 900 kg [24]. It exhibits a circumglobal distribution, inhabiting tropical and temperate waters in all oceans [25], and, similarly to many other sharks from an ecosystem point of view, it plays an essential role as a trophic regulator of ecosystem interaction networks [26]. Tiger sharks exhibit a very varied diet, composed of a wide variety of marine and terrestrial organisms, in addition to carcasses [27,28,29]. The species is ovoviviparous and exhibits a high fecundity, producing between 10 and 82 neonates per posture [30], although it is late to mature, reaching maturity between seven (males) and eight (females) years old [31]. It is important to note that Brazilian coastal records indicate rapid tiger shark growth when compared to other regions worldwide, evidenced by a smaller first maturation size in the northeast coast. (L_50_♀ = 210 cm, L_50_♂ = 237 cm) when compared to the North Atlantic population (L_50_♀ = 320 cm, L_50_♂ = 310 cm) [24,31,32].

Due to a lack of detailed information, capture pressures, and records along the Rio de Janeiro coast, the aim of this study was to expand the tiger shark record database and to improve upon future conservation and management strategies.

## 2. Materials and Methods

### 2.1. Study Area

The coast of Rio de Janeiro comprises 1.160 km, covering a total of 33 municipalities, with its northern limit in the municipality of São Francisco de Itabapoana (21°18′ S) and southern limit in Paraty (23°21′ S). These municipalities represent 40.1% of the state’s territory, where 83% of the state’s population lives. This area exhibits significant economic importance, accounting for 96% of the national offshore oil production and 77% of the national gas production [33]. The coast is divided into four regions (Norte Fluminense, Baixada Litorânea, Metropolitana, and Baía da Ilha Grande) (Figure 1), where a total of 28 artisanal fisher colonies are distributed [34].

All *Galeocerdo cuvier* records obtained off the coast of the state of Rio de Janeiro in this assessment are exhibited in the map depicted in Figure 2. 

### 2.2. Data Sampling

Tiger shark records from the four coastal Rio de Janeiro regions were obtained by direct observation and from fisher colonies/associations, environmental guards, researchers, and published reports. The consulted fisher colonies were the Z13 colony in the metropolitan region of the city of Rio de Janeiro; the Z1 colony in the city of Gargaú, São Francisco de Itabapoana; the Z19 colony in the city of Campos dos Goytacazes (Northern Rio de Janeiro); and the Z17 colony in the city of Angrados Reis (Ilha Grande Bay region). The Tamoios Fishing Association in Cabo Frio (Baixadas Litorâneas region) was also consulted. Consultations were carried out with the fisher colonies/associations and with the environmental guards of each city over the last four years, obtaining reports and some photographic/video records. Lastly, voucher specimens from ichthyological collections of research centers, museums, and universities were recovered from the online databases FishNet 2—www.fishnet2.net (accessed on 12 July 2022), Information System for the Brazilian Biodiversity (SiBBr)—www.ala-bie.sibbr.gov.br (accessed on 12 July 2022) and Rede SpeciesLink—www.specieslink.net (accessed on 12 July 2022) and direct contact with the curators.

## 3. Results

A total of 23 tiger shark records were obtained, with 4 other records identified in articles published in the last 40 years (between 1982 and 2022) along the coast of the state of Rio de Janeiro (Table 1). The coastal areas with the most records were Rio de Janeiro (*N* = 6), followed by São João da Barra (*N* = 5), Campos dos Goytacazes (*N* = 4), Arraial do Cabo (*N* = 3), Cabo Frio, Saquarema, Quissamã, and Angra dos Reis (*N* = 1 each). The northern region of the state exhibited the highest number of records (*N* = 10), followed by the metropolitan lowland regions (*N* = 6), coastal region (*N* = 5), and Ilha Grande Bay (*N* = 1). Two previous records were recovered from voucher specimens deposited at the National Museu (MNRJ) and NUPEM (UFRJ) ichthyological collections. Two neonate specimens, a male and a female, were recovered from the Macaé Municipal Fish Market on 12 December 2017 (NPM 5772). Another individual tiger shark, captured along the Rio de Janeiro coast, was deposited under the catalog number MNRJ 10576.

Four additional records for the city of Rio de Janeiro were obtained from scientific articles by [16,22]. At least six abstracts presented between 1993 and 2018 at scientific events were identified (i.e., Oceanography Weekland Meetings of the Brazilian Society for the Study of Elasmobranchs, among others) mentioning the presence of the species on the coast of Rio de Janeiro, but we did not analyze these data in compliance with journal rules. Only two records of live animals were obtained, whilst the remaining records were from animals that were accidentally caught in artisanal fishing nets. There were four records from artisanal fishers from the northern coast of Rio de Janeiro, with data on catch size and/or weight, photographic evidence, or photographic evidence only [35].

Some morphometric parameters could be obtained from the new records on tiger shark occurrences in this study. The Total Length (TL) of the specimens ranged from 80 to 339.3 cm, with Total Weight (TW) ranging from 23 to 418 kg. Only eight individuals were sexed, comprising two males and six females. The females ranged from 80 cm to 350 cm TL and males from 250 to 335 cm TL. Regarding weight, females ranged from 23 to 300 kg and males, from 200 to 418 kg. The five individuals with the shortest TL (equal to or below 150 cm) were recorded in the city of Rio de Janeiro, and four of these records were obtained between March and September, during the autumn and winter seasons. Another four individuals smaller than 180 cm TL were recorded in the same period in the northern regions of Rio de Janeiro and its coastal lowlands. Another nine individuals (above 200 cm) were recorded between December and March, during the summer season. Several records for juveniles and neonates were obtained, suggesting seasonality patterns and the possibility that the coast of Rio de Janeiro may be a strategic breeding and nesting site for tiger sharks.

Figure 3 displays some of the tiger shark records obtained for the coast of the state of Rio de Janeiro in the present study.

## 4. Discussion

Our results increase the number of records of tiger sharks off the coast of the state of Rio de Janeiro approximately 5-fold and provide information on a possible seasonality pattern concerning the size of the captured/observed animals. Prior to our study, there was some mention of specimens without further details for the cities of Atafona [36], Angra dos Reis, and Rio de Janeiro [37,38]. More recently, four new records for the northern Rio de Janeiro coast were recovered from local fisher knowledge from the last 50 years [35]. Only Araujo et al. [16] presented detailed records of two individuals captured in 2018 and 2019 on Copacabana beach by the Z13 colony, in the metropolitan region of Rio de Janeiro. Although the tiger shark has been reported as among the least abundant shark species on the coast of different Brazilian states, including Rio de Janeiro, we believe that the number of tiger shark records may be higher than the records obtained herein, as we analyzed data from the last 40 years for only about 7% of the 28 colonies and fisher associations in the state. Studies on zooarchaeology in sambaqui shell mounds have identified traces of tiger sharks off the coast of the state on islands near the cities of Angra dos Reis, Cabo Frio, and Guanabara Bay [39,40,41]. Future studies should expand the space–time scale and the number of towns and fishing colonies to be consulted.

The high number of records obtained in northern Rio de Janeiro and the coastal Baixada regions may be due to the presence of the Paraíba do Sul River estuary and a local upwelling system, respectively, together with the fact that tiger sharks display the ability to enter estuarine and freshwater environments to feed. The section near the mouth of the Paraíba do Sul River has, in fact, been identified as presenting the greatest fish richness and diversity during the flooding season when compared to other sections of the same river [42]. However, even though there were no recorded captures of tiger sharks at the mouth of the river, the Paraíba do Sul River plume, which extends to Barra do Açú, results in an increased biomass caused by the greater discharges from the continent, attracting opportunistic species such as tiger sharks. The species may also occur in this region chasing *bonito* and squid shoals that occur during the summer. The upwelling phenomenon of the coastal lowland region, an oceanographic characteristic where deep waters rise continuously resulting in a considerable abundance of nektonic species, makes this one of the most productive fishing areas in the state [43]. The only two live animal observations swimming close to the coast in shallow waters were obtained in this region in 2003 and 2022, both associated with fish schools (Rodrigues and Cumplido pers. obs.).

However, the only record for the Ilha Grande Bay region comprises a 300 cm TL specimen captured 20 years ago on the south side of Ilha Grande. In this region, the consulted young local fisherman did not mention any recent records, and his grandfather reported having caught tiger sharks over 50 years ago. The oldest existing report of a tiger shark in Rio de Janeiro cites the capture of a large specimen in that same region in 1956 [37]. The absence of tiger shark reports among younger fishers has also been identified in the northeast of Brazilian [44]. This suggests that the high industrial fishing pressure caused by the high number of vessels in Ilha Grande Bay (Aximoff pers. obs.) may have caused a decline in the tiger shark population, therefore being responsible for the lack of more recent records. However, we also do not rule out artisanal fishing pressures, as identified in northern Rio de Janeiro [35].

In general, the tiger shark is among the least abundant shark species along the coast of different Brazilian states, such as Recife, Pernambuco [15], Rio de Janeiro itself [16], and near Mel Island in the state of Paraná [17]. This is possibly due to tiger sharks not being the main object of fishing unlike other shark species. This may be taking place in Rio de Janeiro, where few records of tiger shark captures are noted when compared to more abundant species such as sharpnose sharks (*Rhizoprionodon porosus* and *R. lalandii*), angel sharks (*Squatina guggenheim*), and scalloped hammerhead sharks (*Sphyrna lewini*) [16,23]. According to the data from the latest report by the Rio de Janeiro Fisheries Institute Foundation, these same three species presented the highest biomass captured in the state in 2016, at almost 10 tons [45]. According to [35], artisanal fishers in northern Rio de Janeiro consider that the tiger shark used to more common and has suffered a decline in catches over the last few decades.

The records obtained at Copacabana beach, in the metropolitan region of the state, were of two juveniles and one neonate measuring only 89 cm in TL, as these animals are born ranging from about 50 to 70 cm TL [37]. Ref. [16] also identified juveniles of other shark species in Copacabana, suggesting this site as a strategic breeding and nesting site for several species. We suggest that Copacabana beach is strongly influenced by Guanabara Bay, which is considered as one of the most important and productive estuaries in Brazil [46], still exhibiting significant marine and estuarine biodiversity but with significant observed losses for elasmobranch populations [47].

Another study, carried out at Recreio dos Bandeirantes, about 30 km from Copacabana Beach, obtained records of juvenile sharks of several species, indicating that the coast of the metropolitan region of Rio de Janeiro is vital for the reproduction of several elasmobranchs, as indicated by [22], with Guanabara Bay being indicated as a nursery for the butterfly stingray *Gymnura altavela* [48], categorized as Endangered by the International Union for Conservation of Nature. According to [49], some of the criteria employed to identify a nursery area are a high frequency of sharks and the repeated use of the habitat over the years. Two other areas on the coast of the state of São Paulo, southeastern Brazil, are also considered nurseries, thus demonstrating the need for specific conservation measures [40,50].

Regarding the recording period, juveniles have been identified in coastal waters during the autumn and winter seasons, similar to records obtained in the state of Recife [51]. Given that juvenile tiger sharks exhibit high growth rates [24], sexual maturity is reached when these animals reach about 200 cm [37]. At this stage, they then move to deeper waters [52], although they still regularly move to coastal areas to forage [53]. In the Gulf of Mexico, sub-adult and adult tiger sharks achieve significantly higher movement rates than juveniles and inhabit deeper habitats, particularly during the fall and winter seasons [52]. It is considered that the same behavior recorded at the Gulf of Mexico may take place at Guanabara Bay, with tiger sharks that have already surpassed the juvenile stage migrating to new areas to the north and south coasts of the state. In any case, the presence of juvenile tiger sharks demonstrates the importance of the coast of the state of Rio de Janeiro for the species, probably as a feeding and growth area or potentially for reproduction, indicated by the presence of specimens close to the sizes reported for neonates.

The records retrieved herein may comprise the beginning of a baseline for future environmental impact assessments, such as concerning the effects of oil spills and the movements and distribution of tiger sharks off the coast of Rio de Janeiro, which currently accounts for 83% of the offshore oil production in the country [54]. Telemetry monitoring over time throughout the coast of the state is, therefore, paramount to verify the hypotheses in various studies, such as in the study carried out in the oil area of the Gulf of Mexico, where Ajemian et al. [52] investigated the habitat use and the movement patterns of 56 monitored tiger sharks. Telemetry has, in fact, been proven a useful tool to identify variable patterns of the space used by tiger sharks, including residence and migration periods [55].

Finally, the survey carried out here recorded 23 tiger shark occurrences along the coast of Rio de Janeiro with special emphasis on juveniles (eight records), further emphasizing the need to consider the coast of Rio de Janeiro as an especially relevant area for at least part of tiger shark life history. The preservation of Rio de Janeiro’s marine ecosystems significantly depends on the maintenance of its natural top predators, many of which have been considered threatened by the fishing intensity applied to their populations throughout the 20th century [14,35]. Moreover, Guanabara Bay is heavily contaminated by several chemical contaminants due to the inflow of significant amounts of untreated sewage and runoff from landfills, industries, and shipyards [4,56]. In fact, several elasmobranch species captured in or near this area have been reported as containing extremely high levels of both inorganic [57,58,59,60] and organic contaminants ([61], Hauser-Davis, pers. Obs.). This, in turn, comprises a significant threat to their reproduction, development, and survival, especially as many contaminants in the aforementioned studies have been detected in both elasmobranch reproductive and sensory organs.

Furthermore, tiger sharks exhibited a lower reproductive output than expected from the literature for specimens captured along the coast of Hawai’i [62], making juvenile survival and the protection of nursery, feeding, and growing areas potentially critical for conservation strategies, even more so as chemical contamination is of significant concern in juvenile stages [63]. It is also important to note that although pollution has recently been identified as a priority for conservation efforts, especially for elasmobranchs [64,65], data are still scarce for many species, posing a significant knowledge gap for elasmobranch conservation. Additionally, a database such as the one produced herein is the first step toward the design of new research on a species that, although not yet categorized as threatened, is considered key to food web maintenance. This is even more vital in the case of the coast of Rio de Janeiro, where other large predators, many of them also large sharks, similarly face clear declines in their populations.

## 5. Conclusions

This study has expanded tiger shark records 5-fold for Rio de Janeiro, Brazil, reporting 23 new records for this area and providing a baseline for future environmental impact assessments regarding this species. Potential seasonality patterns were noted, and several new records comprised juveniles, which indicates the need to consider the coast of Rio de Janeiro as an especially relevant area for at least part of this species life history and for future conservation strategies.

## Figures and Tables

**Figure 1 animals-12-02774-f001:**
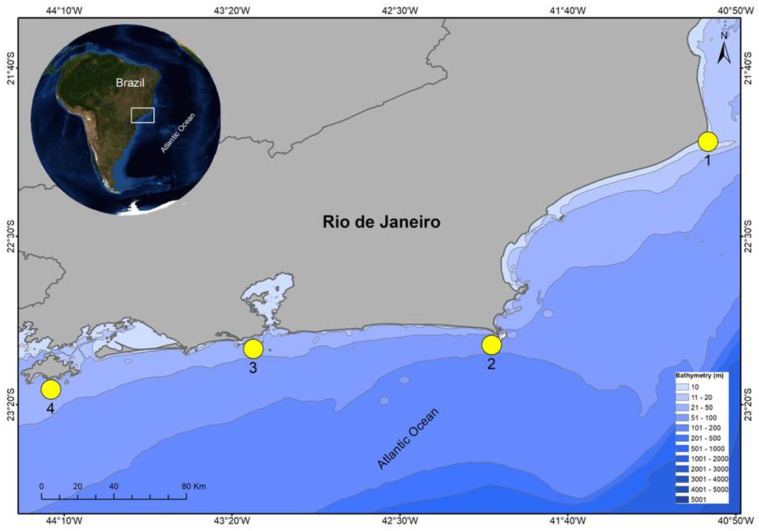
Map indicating the four information sampling regions on *Galeocerdo cuvier* occurrences in the state of Rio de Janeiro, Southeastern Brazil. 1—Norte Fluminense region, 2—Baixada Litorânea region, 3—Metropolitan region and 4—Ilha Grande Bay region.

**Figure 2 animals-12-02774-f002:**
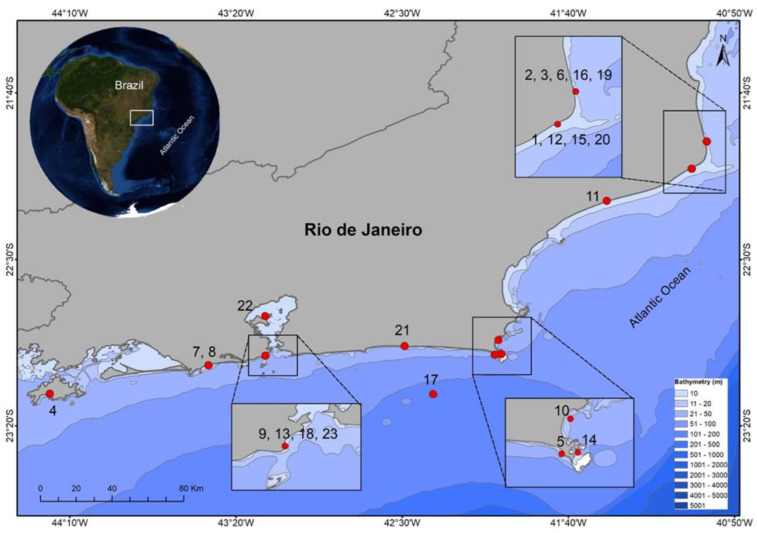
Map of the distribution of *Galeocerdo cuvier* records obtained off the coast of the state of Rio de Janeiro. Each study is identified by the code displayed in Table 1.

**Figure 3 animals-12-02774-f003:**
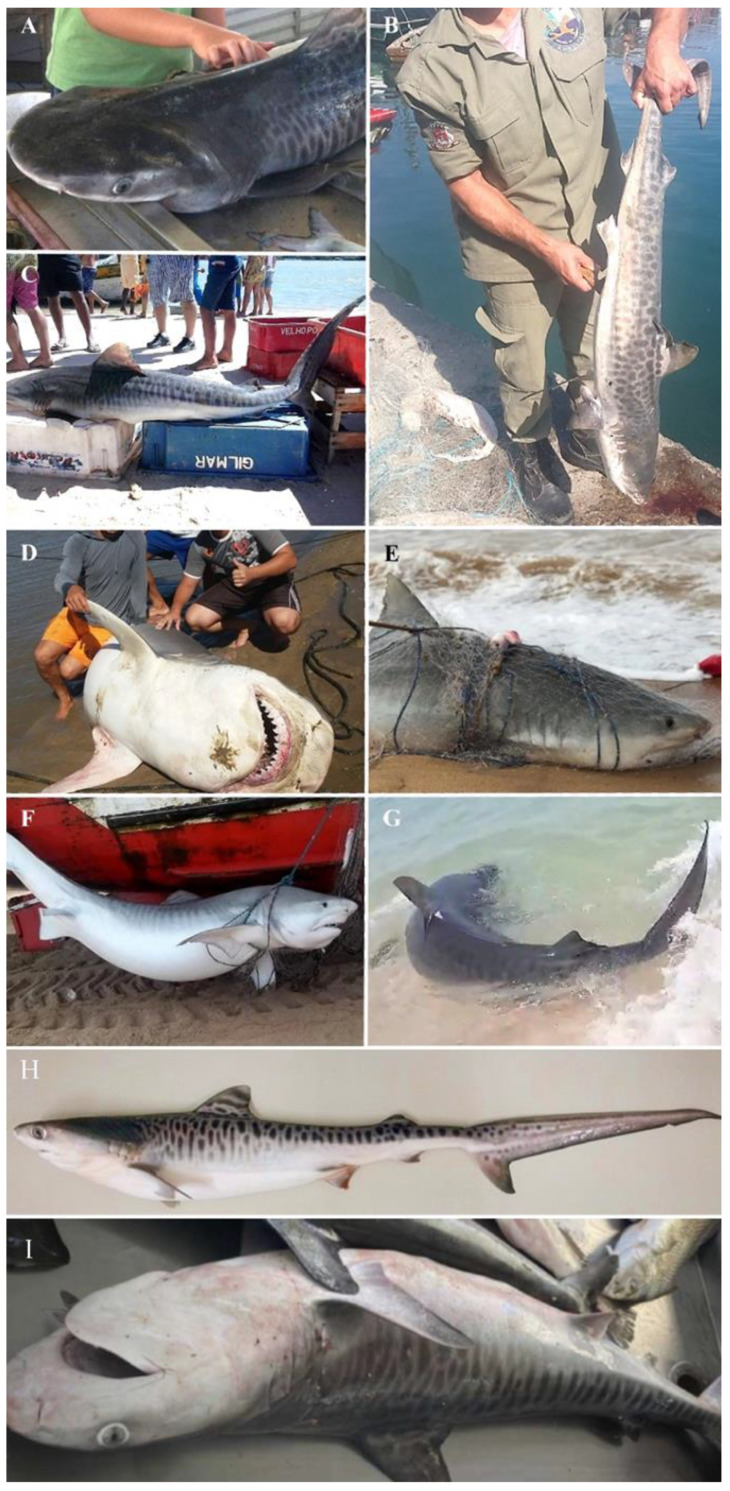
Some of the tiger shark records obtained for the coast of the state of Rio de Janeiro. Photos: (**A**)—Copacabana Beach (2013), (**B**)—Arraial do Cabo (2018), (**C**)—Cabo Frio (2013), (**D**)—Campos dos Goytacazes (2016), (**E**)—São João da Barra (2020), (**F**)—Campos dos Goytacazes (2021), (**G**)—Saquarema (2022), (**H**)—Copacabana Beach (2018), (**I**)—Copacabana Beach (2022).

**Table 1 animals-12-02774-t001:** Morphometric *Galeocerdo cuvier* data for the records obtained off the coast of the state of Rio de Janeiro. Caption: TL = Total Length; TW = Total Weight, * = live animal.** juvenile animal.

Study ID	Month/Year	Sex	TL (cm)	TW (kg)	Source
1	Not informed/1985	-	161	20	Santos et al. (2022a)
2	Not informed/1990	-	170	-	Santos et al. (2022a)
3	Not informed/2000	-	338	220	Current study
4	January/2002	-	300	-	Current study
5	January/2003 *	-	300	-	Current study
6	Not informed/2003	-	393	360	Santos et al. (2022a)
7	Not informed/2004	F	80	-	Silva-Junior et al. (2008)
8	Not informed/2005	-	-	-	Silva-Junior et al. (2008)
9	July/2013	-	150	40	Current study
10	August/2013	F	186	103	Current study
11	February/2016	-	280	220	Current study
12	December/2016	-	250	350	Current study
13	April/2018	F	89	-	Araujo et al. (2020)
14	July/2018	F	160	23	Current study
15	Not informed/2018	-	243	76	Current study
16	February/2019	M	250	200	Current study
17	March/2019	M	335	418	Miranda et al. (2021)
18	September/2019	F	150	-	Araujo et al. (2020)
19	February/2020	-	350	400	Current study
20	December/2021	F	350	300	Current study
21	January/2022 *	-	250	200	Current study
22	June/2022	F	180	75	Current study
23	July/2022 **	F	-	-	Current study

## Data Availability

Not applicable.

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
