# Peer review of "New Occurrences of the Tiger Shark (Galeocerdo cuvier) (Carcharhinidae) off the Coast of Rio de Janeiro, Southeastern Brazil: Seasonality Indications"

_animals, 2022, doi:10.3390/ani12202774_

Round 1
Reviewer 1 Report
Dear Authors,
You have taken up very important issue because of the species. I'm really impressed by your disscusion - not many authors like this chapter and what is more important -know how to write it.
I have some remarks, which I describe line by line:
Line 2 - name of the species with small letters - just tiger shark
7-27 - put a break between number and affiliation
27 - The tiger shark (like in a title) - please do it everywhere. In 72 line it is correct.
41 - don't leave 'a' alone in the end of the line - put it to the next one
43 - Keywords - also in small letters
52, 75, 80, 106, 171, etc. - [3,4] inastead of [3, 4] - change in a whole text
78 - Among the... please delete 'the'
87 - delete 33, it is repetition, we have 33 in 85 line. Moreover, we have These.... so 33 doesn't match here.
91 - (Figure 1) - not in bold.
93 & 97 - please prepare one figure instead of two. Many elements are the same in both figures, moreover there is a big empty place actually between them. Just put yellow dots in figure two and named it as figure one. Actually even figure 2 is not anywhere cited in the text...
Put always a smaller leading between lines - not like under actual Figure 2, but like in Figure 1 (it is correct).
124 - a red coma - why?
142 - **juvenile animal - put a break after ** and before juvenile
145-148 - whole this sentence I would put rather to discussion, not here... it is not your results and you could not used it as you wrote.
154 - 157 - smaller breaks between line, like in actual name of Figure 1.
181 - coastal - a letter in red - why?
282 - lack of chapter called - Conclusions
283-298 - here you need to use only initials instead of full name and surname. It is too long - that's why it is not acceptable.
382, 401, 403, 409 - you should write shorter names oj journals like e.g. in 413 line. Morever , if you do it, in 382 line, it is a hihg probability that '43, 21-31' will be not in separate line.
425 - what does '().' mean? More Authors or is it a mistake?
Good luck.
Reviewer 2 Report
The manuscript must be modified to focus on the species of interest and the information related to the biology and conservation status of the tiger shark, which allows establishing a baseline to further estudies.

Round 2
Reviewer 2 Report
I consider that the manuscript was improved in this new version. However I found some details that should be attended by the authors
Author Response
Responses to Reviewer #2
We have corrected the revised manuscript as requested and inserted all modifications. Rio de Janeiro was maintained instead of RJ, and we have maintained “cities” when requires, as the state, municipality and cities all have the same name, to avoid confusions.
